# Prognosis of Coronary Artery Bypass Grafting in Preschool-Aged Patients with Myocardial Ischemia Due to Giant Aneurysm of Kawasaki Disease

**DOI:** 10.3390/jcm11051421

**Published:** 2022-03-04

**Authors:** Makoto Watanabe, Ryuji Fukazawa, Mitsuhiro Kamisago, Takashi Ohkubo, Masanori Abe, Masami Ochi, Takashi Nitta, Yohsuke Ishii, Shunichi Ogawa, Yasuhiko Itoh

**Affiliations:** 1Department of Pediatrics, Nippon Medical School, Tokyo 113-8603, Japan; s7103@nms.ac.jp (M.W.); kamisago@nms.ac.jp (M.K.); okubo@nishisugamo-child.jp (T.O.); imasha@nifty.com (M.A.); boston@nms.ac.jp (S.O.); yasuhiko@nms.ac.jp (Y.I.); 2Department of Surgery, Cardiovascular Surgery, Nippon Medical School, Tokyo 113-8603, Japan; m-ochi@nms.ac.jp (M.O.); nitta@nms.ac.jp (T.N.); yosuke-i@nms.ac.jp (Y.I.)

**Keywords:** kawasaki disease, CABG, preschool age, graft stenosis, RCA downsizing reconstruction

## Abstract

Coronary artery bypass grafting (CABG) for severe cardiac sequelae of Kawasaki disease (KD) complicated by myocardial ischemia is feasible even in childhood. However, no report has summarized the prognosis of CABG in preschool-aged children. Therefore, we evaluated the outcomes of seven preschool-aged children who underwent CABG for the cardiac sequelae of KD in our hospital. The median age at KD onset and CABG was 36 and 59 months, respectively. The median period from KD onset to CABG was 12 months. The median post-operative observation period was 108 months. CABG between the left internal thoracic artery and left anterior descending artery was performed in all patients. In all patients, postoperative cardiac catheter examination revealed good graft patency and no anastomotic stenosis. Further, pre-operative abnormality of coronary flow reserve returned to normal after CABG. Currently, only one patient is taking warfarin. Regarding school–life management, no patient has exercise limitations, except for one patient who had acute myocardial infarction before CABG. Further, the risk of graft stenosis or occlusion was evaluated in the included patients. However, no accidents have been reported to date, and myocardial ischemia and school–life management have improved. Thus, CABG is an effective treatment in preschool-aged children.

## 1. Introduction

Kawasaki disease (KD) is a vasculitis of unknown cause that occurs in medium-sized systemic arteries. It commonly affects approximately 1-year old infants. Although the cause of KD is unknown, the incidence of coronary artery injury in patients with KD has improved by approximately 5% from the initial 25% owing to progress in therapy [1]. However, the occurrence of myocardial ischemia due to giant coronary aneurysms or advanced coronary artery occlusion is a risk factor for sudden death and is an important problem in pediatric patients. Coronary artery bypass grafting (CABG) in pediatric patients with KD has been established as a treatment for myocardial ischemia since Kitamura et al., performed it with a saphenous vein graft (SVG) in 1976 [2] and the internal thoracic artery (ITA) in 1985 [3]. Takeuchi et al. [4] performed CABG of the right gastroepiploic artery (GEA) in 1990. Currently, SVG is not used as a graft vessel because the patency rate of bypass grafts at postoperative 2 years is 67% for SVG and 93% for ITA [5]. In addition, growth of the ITA graft in association with the growth of children has also been observed [6]. Moreover, patients who underwent CABG with the ITA as a bypass graft had a patency rate of 91% at 15 years after surgery [5]. Currently, the ITA is commonly used as a coronary artery bypass graft in children. The patency rate of coronary artery bypass grafts performed in children <12 years of age since 1994 was 94.4% at 10 years after surgery, which indicates a good prognosis [7]. However, no report has summarized the prognosis of CABG in preschool-aged children. Post-KD hemodynamics are complicated by the presence of dilated and stenotic coronary artery lesions, which are often multi-vessel. Since there are no definite criteria for performing CABG in preschool-aged children, we used coronary fraction reserve (FFR) and coronary flow reserve (CFR) to determine the presence or absence of myocardial ischemia and performed CABG in patients with obvious myocardial ischemia. We evaluated the outcomes in preschool-aged children who underwent CABG for the cardiac sequelae of KD.

## 2. Patients and Methods

### 2.1. Patients

Between 1 January 2000, and 31 October 2014, seven patients underwent CABG at our hospital because of significant myocardial ischemia induced by sequelae of KD. We retrospectively examined their postoperative outcomes. 

We collected data on the age of KD onset, age at the time of CABG, period from KD onset to CABG, period of observation following CABG, medication, and classification of exercise management in the post-CABG follow up. Coronary angiography revealed aneurysm morphology, stenotic lesions, and graft patency after CABG. Left ventricular ejection fraction (LVEF) was evaluated using echocardiography. Myocardial ischemia was evaluated using CFR and FFR. We evaluated FFR and CFR in all patients, and for CFR < 2.0 and/or FFR < 0.8, myocardial ischemia was diagnosed, and CABG was considered. Catheterization was performed under local anesthesia, and sedation using thiamylal sodium was given as needed. If myocardial ischemia is detected by CFR or FFR, then CABG of the left internal thoracic artery–left anterior descending artery (LITA–LAD) should be performed in principle. On the contrary, when ischemia is detected by RCA, we would like to perform CABG if possible, but in many cases, the internal thoracic artery, which can be used as a graft, does not reach the distal portion of the RCA in small children, and the GEA is too thin to be used as a graft; hence, CABG must be given up in many cases. Therefore, in our institution, we perform RCA downsizing reconstruction, with the consent of the patient’s family, to improve ischemia by improving blood flow in the right giant coronary aneurysm [8,9].

### 2.2. Coronary Flow Reserve

The average peak flow velocity (APV) in five consecutive heartbeats was measured using a guide wire with an ultrasonic probe (FloWire XT, Cardiometrics Inc., Mountain View, CA, USA) at rest and after infusing the coronary dilator papaverine hydrochloride. CFR was expressed as the ratio of APV at rest and after papaverine hydrochloride loading (CFR = APV after loading/APV at rest). The CFR is an indicator of the functional severity of myocardial ischemia due to coronary stenosis. The normal value for adults and children is ≥2.0 [10,11]. Papaverine hydrochloride was infused into the left and right coronary arteries at a dose of 0.3 mg/kg (maximum 12 mg) and 0.2 mg/kg (maximum 8 mg), respectively [9].

### 2.3. Fractional Flow Reserve

FFR was calculated as the ratio of pressure at sites distal and proximal to the coronary stenotic lesions, measured using a pressure wire (PressureWire^TM^, RadiMedical, Uppsala, Sweden) after infusion of papaverine hydrochloride. FFR was calculated according to the procedure described in a previous report [12]. FFR suggests ischemia in the perfused myocardial region of the coronary artery and is an indicator of the functional severity of local coronary stenosis [12]. The standard value for children and adults is set to <0.80 [13].

### 2.4. Echocardiography

Left ventricular wall motion and LVEF were evaluated before and after CABG. The most recent data were used as post-CABG data.

### 2.5. Classification of Exercise Management in Post-CABG Follow-Up

The presence and extent of exercise limitation were evaluated based on the School Life Guidance and Management Form, which is generally used to manage school–life activities in Japan [11]. There are five guidance categories, from A to E as follows:A: Home care/hospitalization is required;B: Able to go to school but unable to exercise;C: Able to participate in “light exercise for average students of the same age”;D: Able to participate in “moderate exercise for average students of the same age”; andE: Able to participate in “vigorous exercise for average students of the same age”

### 2.6. Statistics

The age at KD onset, age at CABG, period from KD onset to CABG, period of observation following CABG, and CFR were expressed as medians (upper and lower limits). FFR and LVEF were expressed as mean ± SD. In addition, pre- and post-CABG values were compared using Wilcoxon’s test at a significance level of *p* < 0.05.

### 2.7. Ethics Approval

Patients and their families had the cardiac catheter examination and drug-loaded myocardial scintigraphy, thoroughly explained to them, and the parents of all participants provided written informed consent before the examinations. This study was conducted in accordance with the Declaration of Helsinki, and the protocol was approved by the ethics committee of the Nippon Medical School (No. B-2021-360).

## 3. Results

### 3.1. Patient Characteristics

Of the seven patients, six were male. The median age at KD onset, age at surgery, period from the onset of KD to CABG, and post-CABG observation period was 36 (13, 56), 59 (18, 76), 12 (4, 40), and 108 (37, 134) months, respectively (Table 1).

Giant aneurysms >8 mm in diameter were observed in the left main coronary artery or left anterior descending artery (LAD) in all patients. Six patients had giant aneurysms >20 mm, and three of them had aneurysms >30 mm in length. In five patients, transthoracic echocardiography or intravascular ultrasound revealed thrombi in the giant aneurysms. In three patients, localized stenosis at the outlet of the giant aneurysm was observed on angiography. The coronary lesion sites and cardiac events in each patient are presented in Table 1. Two patients (patients A and G) showed cardiac events due to preoperative myocardial ischemia. Patient A suddenly became ill and their face turned pale during exercise, but the symptoms improved at rest. Patient G had an acute myocardial infarction (AMI) one day after the cardiac catheter examination and underwent urgent CABG.

Regarding the surgical procedure, CABG with a LITA–LAD graft was performed in all cases, and no patient underwent CABG to the left circumflex coronary artery (LCx) or right coronary artery (RCA) (Table 1). RCA downsizing reconstruction was performed in three cases (Cases E, F, and G). No perioperative mortality was observed, and all patients were discharged without symptoms after CABG. No cardiac events were observed during the follow-up.

### 3.2. Evaluation of CFR, FFR and Stenosis in the Bypass Graft

A comparison of CFR and FFR before and after CABG and the presence or absence of bypass anastomotic stenosis is shown in Table 2. CFR and FFR were evaluated before CABG and 3–4 months after CABG, and there was no occlusion of the left coronary artery aneurysm or downsizing of the right coronary artery aneurysm at that time. In LAD, the CRF values before CABG were abnormal (<2.0) in all patients but normalized after CABG (1.41 ± 0.30 vs. 2.64 ± 0.35, *p* = 0.0021). FFR was measured in five patients before CABG. A total of three patients (B, C and E) with stenosis at the outlet of the giant aneurysm showed a decreased FFR. The FFR normalized after CABG. No graft stenosis was observed in any patient. In RCA downsizing reconstruction, the CFR values reduced in three cases (cases E, F, and G). However, FFR was not reduced in all patients. The CFR values normalized after downsizing construction.

### 3.3. LVEF, Left Ventricular Wall Motion Abnormalities

There was no significant improvement in the LVEF after surgery (66.1 ± 3.3, vs. 65.7 ± 10.3, *p* = 0.6674) (Table 2). Wall motion abnormalities were observed in all patients before CABG. Postoperatively, obvious wall-motion abnormalities were identified in only two patients. In patient B, abnormal motion of the anterior septum due to asymptomatic myocardial infarction caused by acute-phase aneurysms of the anterior descending branch was observed. In patient G, AMI occurred one day after the pre-CABG catheter examination, and a decrease in LVEF was observed before CABG. In these two patients, LVEF continued to decrease after CABG.

### 3.4. Latest CAG Findings

The median age at the latest post-operative cardiac catheterization was 59 months since CABG. The CABG graft patency was satisfactory (Table 3). The aneurysms in the LADs were occluded except for those in Case B. The patient in Case B had a 75% localized stenosis in the distal part of the aneurysm and no flow competition with the graft flow. The patient in Case F had a large left main coronary trunk (LMT) aneurysm and the LAD was occluded, but blood flow to the LCx was maintained through the aneurysm and there was no stenosis. In case G, which showed a large aneurysm of the LMT, the aneurysm was occluded, but blood flow to the LCx periphery was observed as segmental stenosis.

In Cases C and D with RCA occlusion, segmental stenosis occurred and blood flow to the RCA periphery was observed.

In Cases E, F, and G, where RCA aneurysm downsizing construction was performed, aneurysms remained, but contrast flow was smoother and CFR improved (Figure 1 and Figure 2).

### 3.5. Post-CABG Medication and Classification of Exercise Management

All patients were taking warfarin before CABG, but only one patient (patient F) required warfarin after CABG (Table 3). This patient had a giant aneurysm proximal to the LAD and LCx, and underwent CABG with a LITA–LAD graft. However, the LCx still arose from the giant aneurysm; therefore, warfarin had to be continued. Aspirin was prescribed to all patients. Candesartan was prescribed to four patients who were expected to have inhibited vascular remodeling. In patient G, enalapril and carvedilol were prescribed for the reduced cardiac function induced by AMI.

After CABG, there was no exercise limitation in school and daily life, except in patient G, who had reduced left ventricular function due to AMI. Except patient G, all patients were classified into category E in the School Life Guidance and Management Form. Patient F, who needed continuation of warfarin, was restricted from contact sports such as soccer or rugby.

## 4. Discussion

All preschool-aged children who underwent CABG with the ITA at our hospital showed improvement in ischemia in the target coronary artery region. All patients were taking warfarin before CABG, but only one patient required warfarin post-operatively. Except one patient who showed reduced left ventricular function due to AMI before CABG, all patients could participate in all school activities, including strenuous athletic activities. In addition, no case of bypass anastomotic stenosis has been observed thus far, and neither repeat surgery, nor intervention has been performed. CABG is considered an effective and satisfactory procedure in preschool-aged children with KD sequalae. 

Because the blood flowing into giant aneurysms becomes turbulent, it loses its initial energy, resulting in a decrease in the perfusion pressure. Thus, giant aneurysms exhibit the same function as stenotic lesions due to hydrodynamics [14]. Therefore, it is possible that the FFR reflecting the stenosis of the epicardial artery showed an abnormal value even with a 75% stenosis (Case B). Moreover, stenosis is likely to occur at both the inlet and outlet of the aneurysm [15]. Even if the stenosis is released by catheter intervention, it is highly likely to recur. Therefore, CABG is a promising choice, especially when choosing a treatment for ischemia associated with giant aneurysms [8]. In general, the indications include (1) significant stenosis of the left main trunk; (2) multivessel disease; (3) stenosis in the proximal portion of the LAD; and (4) jeopardized collaterals [13]. Furthermore, in our institution, we actively perform CABG if (5) giant aneurysms are hemodynamically similar to highly occlusive lesions, and there is evidence of ischemia distally; (6) uncontrolled thrombus formation occurs as a result of medical treatment of intra-aneurysmal thrombosis; (7) severe stenosis and myocardial ischemia in the distal part of the aneurysm, and (8) the LAD is completely occluded and variability is observed in the ischemic region [9]. Among those criteria, we considered the existence of ischemia as the most important one. Myocardial ischemia may put children at risk of sudden death and cardiac events for the rest of their lives. Both CFR and FFR are indicators of coronary blood flow. FFR is primarily an index for evaluating the degree of epicardial artery stenosis, but it can also be a surrogate index for evaluating coronary blood flow [16]. By contrast, CFR is an index that reflects not only blood flow in the epicardial artery but also blood flow in the peripheral circulation [17,18]. In the present study, CFR was decreased in all patients, suggesting myocardial ischemia, and CFR increased significantly after CABG. Echocardiographic wall motion abnormalities were also improved. In patients with stenosis at the outlet of giant aneurysm, not only CFR but also FFR decreased. We believe that such an overall increase in peripheral blood flow by CABG may reduce the risk of graft occlusion. Hence, it is important to evaluate both FFR and CFR indices to assess myocardial ischemia in complex vasculatures of KD sequalae.

In pediatric patients, there is a risk of postoperative graft stenosis or graft occlusion because of the small vessels and body size. However, no such complications have been reported to date in any case. Tsuda et al., reported a high patency rate of the ITA as a graft for CABG in KD [5,7]. The patency rates of the SVG at 1, 5, and 15 years after surgery were 84%, 65%, and 53%, respectively, whereas those of the ITA were 94%, 79%, and 73%, respectively. When limited to patients aged ≥12 years, the ITA patency rates were 95%, 91%, and 91%, respectively. On the other hand, with regard to the GEA, the patency rates at 1 and 5 years after surgery were 72% and 43%, respectively. Thus, the ITA is used for all CABGs performed in our hospital. In addition, no CABG with the Y-graft technique or GEA is performed because the patency rate is low in the pediatric population. None of the patients underwent CABG to the RCA. One reason is that patients’ body size was so small that the length of the right ITA or size of the GEA was limited [8]. Blood supply to the region of the RCA distribution can be expected via collateral vessels from the LAD or LCx, and recanalization of the RCA can be expected after occlusion. Thus, CABG to the RCA was not performed. In addition, our hospital recommends the following procedures for CABG in children: (1) in principle, revascularization should be performed using only in situ (not free but pedicled) ITAs; and (2) sequential grafting of the left ITA (i.e., LITA to LAD and diagonal branch) is sometimes feasible, if needed. However, we do not recommend CABG using the Y-graft technique or GEA; (3) CABG is performed on the arrested heart using cardiopulmonary bypass to obtain reliable grafting. 

RCA downsizing construction is a technique designed to improve ischemia in the RCA region in children with a small stature who are not candidates for CABG. By reducing the diameter of a large coronary aneurysm, blood flow can be changed from turbulent to laminar, thereby reducing the energy loss due to blood flow and improving blood flow to the periphery. In addition, it is expected to prevent thrombus formation and to improve endothelial function by improving shear stress on the vessel wall. In our experience, CFR has been improved, laminar blood flow has been restored, and warfarin has been discontinued in some cases (manuscript in preparation). We hope to accumulate more cases related to this surgery in the future.

In recent years, the risk of early atherosclerosis has been cited as a problem in the remote stage of KD, and statin therapy in the remote stage has been recommended [13,19]. Although direct evidence of the relationship between KD and atherosclerosis is lacking, we attempted to save one of the ITAs considering the risk of future acute coronary syndromes.

### Study Limitations

This was a single-institution study, which included only seven patients. In future, it is necessary to conduct a nationwide survey to determine whether CABG surgery in young children is safe and has an excellent prognosis. Further, the median post-operative observation period was 108 months (9 years). Careful follow up of the patients is necessary to assess long-term outcomes.

In the case of pediatric catheterization, sedation is often required for safety reasons, and the depth of sedation may affect coronary blood flow, which may, in turn, affect the evaluation of FFR and CFR. At our institution, sedation is kept to a minimum level by a specialist so that it does not cause vital changes, and the examination is performed under spontaneous breathing without the need for oxygen inhalation. For this reason, we believe that FFR and CFR do not affect the determination of myocardial ischemia, but this has not yet been proven with certainty.

## 5. Conclusions

CABG for cardiac sequelae of KD improved myocardial ischemia and school life, including athletic activities, in preschool-aged children. Thus, CABG is considered to be effective even in preschool-aged children.

## Figures and Tables

**Figure 1 jcm-11-01421-f001:**
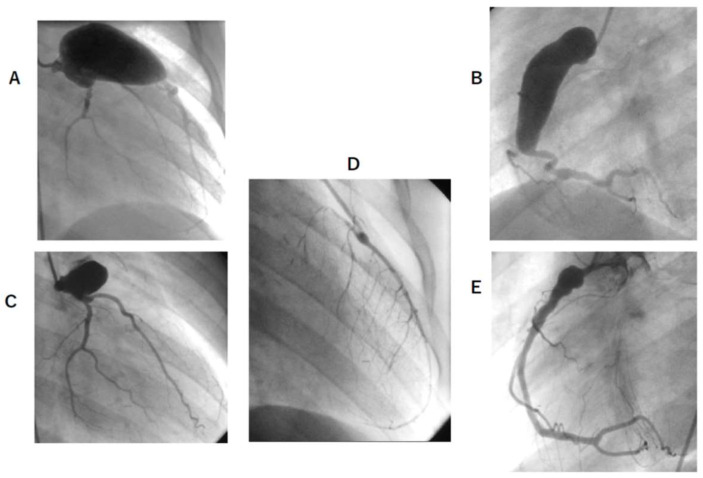
CAG of patient E before and after 92 months of CABG and RCA downsizing reconstruction. (**A**,**B**) are before CABG. And (**C**,**D**,**E**) are after 92 months of CABG. (**A**), LCA angiography. Giant aneurysm 19.0 × 39 mm were located. No stenotic lesion was observed; (**B**), RCA angiography. Giant aneurysm RCA: 12.0 × 42 mm was observed; (**C**), LCA angiography. LAD was occluded. LCx arose from aneurysm; (**D**), LITA angiography. LITA–LAD bypass patency was good. (**E**); RCA angiography. The shape of RCA became good after downsizing reconstruction, remaining 8.8 × 12 mm dilation at proximal. Wash-out of the contrast medium became smooth. CAG, coronary angiogram; CABG, coronary artery bypass grafting; LCA, left coronary artery; RCA, right coronary artery; LAD, left anterior descending; LCx, left circumflex; LITA, left intimal thoracic artery.

**Figure 2 jcm-11-01421-f002:**
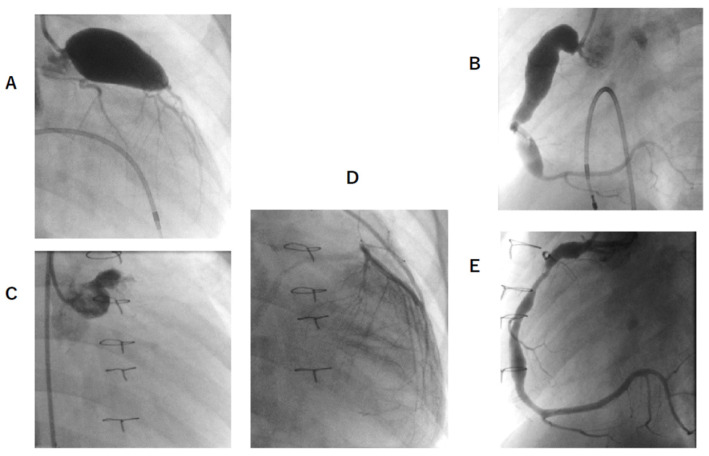
CAG of patient F before and after 18 months of CABG and RCA downsizing reconstruction. (**A**,**B**) are before CABG. And (**C**,**D**,**E**) are after 18 months of CABG. (**A**), LCA angiography. Giant aneurysm 16.5 × 31 mm were located. No stenotic lesion was observed; (**B**), RCA angiography. Giant aneurysm RCA, sized 10.0 × 37 mm, and medium aneurysm, sized 7.0 × 17 mm, were observed; (**C**), LCA angiography. LCA was occluded. LCx showed segmental stenosis after this; (**D**), LITA angiography. LITA–LAD bypass patency was good. (**E**); RCA angiography. The shape of RCA became good after downsizing reconstruction. Wash-out of the contrast medium became smooth.

**Table 1 jcm-11-01421-t001:** Patient background and surgical procedure.

Patient	Sex	Age at Onset (Months)	Age at CABG (Months)	Period from Onset to CABG (Months)	Post-Surgery Observation Period (Months)	Coronary AN (Diameter × Length)	Stenosis	Thrombus	Event	Surgical Procedure
A	Female	32	51	19	134	LAD: 12.0 × 28 mm RCA: 7.0 × 12 mm	No	Yes	Angina	LITA–LAD
B	Male	13	38	25	108	LAD: 10.0 × 20 mm	75% LS at AN outlet	Yes	No	LITA–LAD
C	Male	36	76	40	110	LAD: 8.5 × 12 mm RCA: Occluded	90% LS at AN outlet	No	No	LITA–LAD
D	Male	14	18	4	46	LAD: 17.5 × 40 mm LCx: 10.0 × 15 mm RCA: Occluded	No	Yes	No	LITA–LAD
E	Male	54	59	5	37	LAD: 15.2 × 23 mm RCA: 7.5 × 17 mm	90% LS at AN outlet	No	No	LITA–LAD RCA down-sizing
F	Male	50	62	12	115	LMT: 19.0 × 39 mm RCA: 12.0 × 42 mm	No	Yes	No	LITA–LAD RCA down-sizing
G	Male	56	65	9	42	LMT: 16.5 × 31 mm RCA: 10.0 × 37 mm, 7.0 × 17 mm	No	Yes	AMI	LITA–LAD RCA down-sizing
		36 (13, 56)	59 (18, 76)	12 (4, 40)	108 (37, 134)					

CABG, coronary artery bypass grafting; LITA, left internal thoracic artery; LAD, left anterior descending artery; RCA, right coronary artery; LMT, left main coronary trunk; AMI, acute myocardial infarction; LS, localized stenosis; AN, aneurysm.

**Table 2 jcm-11-01421-t002:** Comparison of CFR, FFR, LVEF, and LV wall motion before and after CABG and the presence or absence of bypass anastomotic stenosis.

Patient	CFR		FFR		Anastomotic Site Stenosis	LVEF (%)		LV Wall Motion Abnormalities	
	Pre-CABG	Post-CABG	Pre-CABG	Post-CABG		Pre-CABG	Post-CABG	Pre-CABG	Post-CABG
A	LAD: 1.2 RCA: 2.3	LAD: 3.0 RCA: 2.6	LAD: 0.86 RCA: 0.88	LAD: 0.87 RCA: 0.88	No	64	70	Anterior septum	No
B	LAD: 1.9	LAD: 2.7	LAD: 0.56	LAD: 0.81	No	62	60	Anterior septum	Anterior septum
C	LAD: 1.6	LAD: 3.1	LAD: 0.59	LAD: 0.82	No	68	75	Septum; Apical region	No
D	LAD: 1.0	LAD: 2.4	LAD: 0.85		No	64	63	Anterior septum; Apical region	No
E	LAD: 1.6 RCA: 1.3	LAD: 2.7 RCA: 3.4	LAD: 0.65 RCA: 0.95	LAD: 0.94 RCA: 0.85	No	70	74	Anterior septum; Apical region	No
F	LAD: 1.3 RCA: 1.2	LAD: 2.5 RCA: 2.5	LAD: 0.94 RCA: 0.93	LAD: 0.96 RCA: 0.85	No	69	72	Anterior-lateral wall; Septum	No
G	LAD: 1.3 RCA: 1.3	LAD: 2.1 RCA: 2.6		LAD: 0.84 RCA: 0.94	No	AMI occurred one day after CAG	46	Anterior septum; Apical region	Diffuse decrease in wall motion
Mean	LAD: 1.41 ± 0.30 RCA: 1.53 ± 0.52	LAD: 2.64 ± 0.35 * RCA: 2.78 ± 0.42 **	LAD: 0.7 ± 0.16 RCA: 0.9 ± 0.03	LAD: 0.86 ± 0.05 ^†^ RCA: 0.88 ± 0.04 ^††^		66.1 ± 3.3	65.7 ± 10.3		

The CFR and FFR were evaluated at LAD segment 7 or RCA segment 3. * *p* = 0.0021, ** *p* = 0.0284, ^†^
*p* = 0.4225, ^††^
*p* = 0.2801, ^¶^
*p* = 0.6674. CFR, coronary flow reserve; FFR, fraction flow reserve; LAD, left anterior descending artery; CABG, coronary artery bypass grafting. LAD, left anterior descending artery; RCA, right coronary artery; LV, left ventricle; LVEF, left ventricular ejection fraction; CABG, coronary artery bypass grafting; CAG, coronary angiography. Patient G had AMI a day after the cardiac catheter examination and underwent urgent CABG.

**Table 3 jcm-11-01421-t003:** The latest CAG findings; Current medications and symptoms; Exercise limitation.

Patient	Latest CAG Findings	Current Medications	Symptoms	Exercise Limitation
A	LITA–LAD patency good LAD AN occluded RCA AN: 6.0 *×* 12 mm	Aspirin	No	E
B	LITA–LAD patency good LAD AN outlet 75% LS	Aspirin, Candesartan	No	E
C	LITA–LAD patency good LAD AN occluded RCA segmental stenosis	Aspirin, Candesartan	No	E
D	LITA–LAD patency good LAD AN occluded LCx AN regressed RCA segmental stenosis	Aspirin, Candesartan	No	E
E	LITA–LAD patency good LAD AN occluded RCA AN: 6.2 *×* 15 mm	Aspirin, Candesartan	No	E
F	LITA–LAD patency good LAD AN occluded LCx arise from AN RCA AN: 8.8 *×* 12 mm	Warfarin, Aspirin,	No	E (Restricted from contact sports)
G	LITA–LAD patency good LMT AN occluded LCx segmental stenosis RCA AN 5.4 *×* 12, 4.2 *×* 11, and 5.2 *×* 11 mm	Aspirin, Enalapril, Carvedilol	No	D

Exercise Limitation: A: Home care/hospitalization is required; B: Able to go to school but unable to exercise; C: Able to participate in “light exercise for average students of the same age”; D: Able to participate in “moderate exercise for average students of the same age”; and E: Able to participate in “vigorous exercise for average students of the same age” CABG, coronary artery bypass grafting; LITA, left internal thoracic artery; LAD, left anterior descending artery; RCA, right coronary artery; LMT, left main coronary trunk; LS, localized stenosis; AN, aneurysm.

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
