# Peer review of "Prognosis of Coronary Artery Bypass Grafting in Preschool-Aged Patients with Myocardial Ischemia Due to Giant Aneurysm of Kawasaki Disease"

_jcm, 2022, doi:10.3390/jcm11051421_

Round 1
Reviewer 1 Report
CABG is rare and challenging in children under 5 years old with Kawasaki disease and giant coronary artery aneurysm. This is an article of great clinical value. Here are some questions and details to be delineated:
1. It is suggested that tables 2, 3 and 4 be combined into one table for easy referring;
2. The surgical indications of giant coronary artery aneurysm are the key point in clinic. This paper takes the myocardial ischemia detected by CFR and FFR as the criterion of CABG surgery. It is challenging to get accurate CFR and FFR in cath lab for preschool children. It is suggested that the authors describe it in the method and discussion part;
3.For giant coronary artery aneurysm without occlusion or severe stenosis, aneurysm angioplasty was carried out in some centers. CABG was performed for such patients in this paper, which is suggested to be discussed for these two methods;
4.From the table 2, the patient A / D / F /G have no coronary stenosis. Does the postoperative angiography show competitive blood flow? And how about the original giant coronary artery aneurysm shown by postoperative angiography? May more pictures be shown including preoperative and postoperative angiogram pictures.
5. In patient F, LCX arose from the giant aneurysm, so warfarin was required after-CABG. However, as shown in Table 2, the length of the giant aneurysm of patient A/ D / F / G in LMT was 28-40mm. In such younger patients, the length of LMT should not longer than 28-40mm, so the LCX should arise from these giant aneurysms. Why warfarin was stopped for these patients? Please explain or give more images.
6. In Table 5, why did patient F take Dipyridamole?
Author Response
To Reviewer 1
We thank you for providing useful comments. We have made the necessary corrections in accordance with your comments. These comments have greatly improved the quality of our manuscript. Our point-by-point responses are shown below.
- It is suggested that tables 2,3 and 4 be combined into one table for easy referring.
Response: Thank you for your suggestion. We have merged Tables 1 and 2 as well as Tables 3 and 4.
- The surgical indications of giant coronary artery aneurysm are the key point in clinic. This paper takes the myocardial ischemia detected by CFR and FFR as the criterion of CABG surgery. It is challenging to get accurate CFR and FFR in cath lab for preschool children. It is suggested that the authors describe it in the method and discussion part.
Response: Thank you for your comment. We have added the aforementioned sentence in the Methods (Lines 89-91).
“Catheterization was performed under local anesthesia, and sedation using thiamylal sodium was given as needed.”
In the study limitations section, we have added the following text (Lines 405-412):
“In the case of pediatric catheterization, sedation is often required for safety reasons, and the depth of sedation may affect coronary blood flow, which may, in turn, affect the evaluation of FFR and CFR. At our institution, sedation is kept to a minimum level by a specialist so that it does not cause vital changes, and the examination is performed under spontaneous breathing without the need for oxygen inhalation. For this reason, we believe that FFR and CFR do not affect the determination of myocardial ischemia, but this has not yet been proven with certainty.”
- For giant coronary artery aneurysm without occlusion or severe stenosis, aneurysm angioplasty was carried out in some centers. CABG was performed for such patients in this paper, which is suggested to be discussed for these two methods.
Response: Thank you for your comment. Perhaps you refer to the downsizing reconstruction of a giant aneurysm, which is the main procedure performed at our institution.
Although we focused only on CABG in this study, we performed downsizing reconstruction of the right coronary artery in Cases E, F and G. We have added a description of this.
We have added the sentence below (Lines 91-102)
“If myocardial ischemia is detected by CFR or FFR, then CABG of the left internal thoracic artery–left anterior descending artery (LITA-LAD) should be performed in principle. On the contrary, when ischemia is detected by RCA, we would like to perform CABG if possible, but in many cases, the internal thoracic artery, which can be used as a graft, does not reach the distal portion of the RCA in small children, and the GEA is too thin to be used as a graft; hence, CABG must be given up in many cases. Therefore, in our institution, we perform RCA downsizing reconstruction with the consent of the patient's family to improve ischemia by improving blood flow in the right giant coronary aneurysm.”
We have also added the following content(Lines 213-216):
“In RCA downsizing reconstruction, the CFR values reduced in three cases (Cases E, F, and G). However, FFR was not reduced in all patients. The CFR values normalized after downsizing construction.”
“CFR and FFR were evaluated before CABG and 3–4 months after CABG, and there was no occlusion of the left coronary artery aneurysm or downsizing of the right coronary artery aneurysm at that time.” (Lines 200-203)
“RCA downsizing construction is a technique designed to improve ischemia in the RCA region in children with a small stature who are not candidates for CABG. By reducing the diameter of a large coronary aneurysm, blood flow can be changed from turbulent to laminar, thereby reducing the energy loss due to blood flow and improving blood flow to the periphery. In addition, it is expected to prevent thrombus formation and to improve endothelial function by improving shear stress on the vessel wall. In our experience, CFR has been improved, laminar blood flow has been restored, and warfarin has been discontinued in some cases (during manuscript in preparation). We hope to accumulate more cases related to this surgery in the future.” (Lines 380-390)
Response: We have added a description about RCA in Table 1. We have also mentioned about CFR and FFR of RCA in Table 3. Finally, we have added the latest CAG findings in Table 3.
- From the Table 2, the patient A/D/F/G have no coronary stenosis. Does the postoperative angiography show competitive blood flow? And how about the original giant coronary artery aneurysm shown by postoperative angiography? Many more pictures be shown including preoperative and postoperative angiogram pictures.
Response: Thank you for pointing out these important aspects. All aneurysms of LAD, except those in Case B, were occluded. We have added the latest CAG findings in Results and Table 3, as well as images (Figures 1 and 2).
We have added the following sentence in the Results (Lines 242-257):
3.4 Latest CAG findings
“The median age at the latest postoperative cardiac catheterization was 59 months since CABG. The CABG graft patency was satisfactory (Table 3). The aneurysms in the LADs were occluded except for those in Case B. The patient in Case B had a 75% localized stenosis in the distal part of the aneurysm and no flow competition with the graft flow. The patient in Case F had a large left main coronary trunk (LMT) aneurysm and the LAD was occluded, but blood flow to the LCx was maintained through the aneurysm and there was no stenosis. In case G, which showed a large aneurysm of the LMT, the aneurysm was occluded, but blood flow to the LCx periphery was observed as segmental stenosis.
In Cases C and D with RCA occlusion, segmental stenosis occurred and blood flow to the RCA periphery was observed.
In Cases E, F, and G, where RCA aneurysm downsizing construction was performed, aneurysms remained, but contrast flow was smoother and CFR improved (Figures 1 and 2).”
- In patient F, LCx arose from the giant aneurysm, so warfarin was required after-CABG. However, as shown in Table 2, the length of the giant aneurysm of patient A/D/F/G in LMT was 28-40mm. In such younger patients, the length of LMT should be longer than 28-40mm, so the LCx should arise from these giant aneurysms. Why was warfarin stopped for these patients? Please explain or give mor images.
Response: Please see our response to question 4.
- In Table 5, why did patient F take Dipyridamole?
Response: Thank you for pointing this out. We stopped prescribing dipyridamole several years after CABG.

Reviewer 2 Report
The paper by Watanabe et al. reported the long-term efficacy of CABG in pre-school aged patients with Kawasaki disease. Of note, all patients underwent myocardial ischemia assessment by CFR and FFR before revascularization. Even if the study population consists of only seven patients and the mean postoperative follow-up was only nine years, the main findings of this study are remarkable because of the lack of data in this setting and the few cases previously described; therefore, as stated by the same authors, national and international registries should be strongly encouraged. The paper is well written, methods and results are appropriately reported. Some points to potentially improve the paper:
-All patients underwent LITA-LAD bypass; the authors clearly described in the discussion the reasons to not perform arterial and venous bypass in other coronary vessels such as the right coronary artery. However, did they investigate ischemia also in this coronary branch or only on LM-LAD? No data regarding RCA were reported in Table 3.
-In patients with aneurysms/stenosis in the right coronary artery, they still prescribe aspirin. Why not oral anticoagulation? Additionally, in those with potential evidence of ischemia in this myocardial territory, beta-blockers might be considered to prevent ischemia and arrhythmias. Please comment.
- Moreover, they correctly mentioned statins and their potential role in the Kawasaki disease in the discussion. Probably they could discuss why these drugs were not prescribed in the present population.
- On page 5, line 184, the authors reported: “The median age at the time of postoperative cardiac catheterization was 59 months”. In table 1, this was the median age at the time of CABG, whereas the interval between CABG and angiographic follow-up was 108 months. Probably something is wrong. Please check the data in the text and table or clarify. Thanks.
- The abstract might be shortened, specifically in the last part “Further, preoperative abnormality of coronary flow reserve returned to normal after CABG. Currently, only one patient is taking warfarin. Regarding school-life management, no patient has exercise limitations, except for one patient who had acute myocardial infarction before CABG. Further, the risk of graft stenosis or occlusion was evaluated in the included patients. However, no accidents have been reported to date, and myocardial ischemia and school-life management have improved”.
Author Response
To Reviewer 2
We sincerely thank you for the useful comments. We have corrected the points you had commented on. These suggestions have largely improved the quality of our paper. Our point-by-point responses are shown below:
- All patients underwent LITA-LAD bypass; the authors clearly described in the discussion the reasons not to perform arterial and venous bypass in other coronary vessels such as the right coronary artery. However, did they investigate ischemia also in this coronary branch or only on LM-LAD? No data regarding RCA were reported in Table 3.
Response: Thank you for the suggestion on the evaluation of RCA for ischemia. We would like to use CABG with ITA as a general rule even in cases where ischemia is found in RCA in adults. However, we have mentioned that it may be technically difficult to use ITA or GEA in children with a small stature. In our institution, we perform downsizing reconstruction of giant aneurysms of the right coronary artery to improve blood flow in cases of giant aneurysms of the right coronary artery with ischemia, if their parents' consent is obtained.
We have added the sentence shown below (Lines 91-102)
“If myocardial ischemia is detected by CFR or FFR, then CABG of the left internal thoracic artery–left anterior descending artery (LITA-LAD) should be performed in principle. On the contrary, when ischemia is detected by RCA, we would like to perform CABG if possible, but in many cases, the internal thoracic artery, which can be used as a graft, does not reach the distal portion of the RCA in small children, and the GEA is too thin to be used as a graft; hence, CABG must be given up in many cases. Therefore, in our institution, we perform RCA downsizing reconstruction with the consent of the patient's family to improve ischemia by improving blood flow in the right giant coronary aneurysm.”
We also added the sentence shown below (Lines 213-216):
“In RCA downsizing reconstruction, the CFR values reduced in three cases (Cases E, F, and G). However, FFR was not reduced in all patients. The CFR values normalized after downsizing construction.”
“CFR and FFR were evaluated before CABG and 3–4 months after CABG, and there was no occlusion of the left coronary artery aneurysm or downsizing of the right coronary artery aneurysm at that time.” (Lines 200-203)
“RCA downsizing construction is a technique designed to improve ischemia in the RCA region in children with a small stature who are not candidates for CABG. By reducing the diameter of a large coronary aneurysm, blood flow can be changed from turbulent to laminar, thereby reducing the energy loss due to blood flow and improving blood flow to the periphery. In addition, it is expected to prevent thrombus formation and to improve endothelial function by improving shear stress on the vessel wall. In our experience, CFR has been improved, laminar blood flow has been restored, and warfarin has been discontinued in some cases (during manuscript in preparation). We hope to accumulate more cases related to this surgery in the future.” (Lines 380-390)
We have added a description about RCA in Table 1. We have also mentioned about CFR and FFR of RCA in Table 3. Finally, we have added the latest CAG findings in Table 3.
- In patients with aneurysm/stenosis in the right coronary artery, they still prescribe aspirin. Why not oral anticoagulation? Additionally, in those with potential evidence of ischemia in this myocardial territory, beta-blockers might be considered to prevent ischemia and arrhythmias. Please comment.
Response: Thank you for your comment. Please refer to the response to the previous question regarding the right coronary artery. Additionally, please refer to Figures 1 and 2 to see the final result of the right coronary artery downsizing reconstruction. Except for Case F, who still has a large aneurysm, oral anticoagulants are considered unnecessary and only antiplatelet agents are used. We have also added the CFR and FFR of the right coronary artery to Table 2. At this point, we believe that ischemia has improved.
We have added the latest CAG findings in Lines 236-252 and Table 3.
3.4 Latest CAG findings
“The median age at the latest postoperative cardiac catheterization was 59 months since CABG. The CABG graft patency was satisfactory (Table 3). The aneurysms in the LADs were occluded except for those in Case B. The patient in Case B had a 75% localized stenosis in the distal part of the aneurysm and no flow competition with the graft flow. The patient in Case F had a large left main coronary trunk (LMT) aneurysm and the LAD was occluded, but blood flow to the LCx was maintained through the aneurysm and there was no stenosis. In case G, which showed a large aneurysm of the LMT, the aneurysm was occluded, but blood flow to the LCx periphery was observed as segmental stenosis.
In Cases C and D with RCA occlusion, segmental stenosis occurred and blood flow to the RCA periphery was observed.
In Cases E, F, and G, where RCA aneurysm downsizing construction was performed, aneurysms remained, but contrast flow was smoother and CFR improved (Figures 1 and 2).”
- Moreover, they correctly mentioned statins and their potential role in the Kawasaki disease in the discussion. Probably they could discuss thy these drugs were not prescribed in the present population.
Response: Thank you for your suggestion. As you mentioned, statins are expected to be used for the treatment of the remote phase of Kawasaki disease in the future. In Japan, however, statins are not yet covered by insurance and should be used with caution in children. The Japanese Guidelines for the Treatment of Kawasaki Disease in the Remote Phase, revised in 2020, also include this as recommendations but have not mentioned the specific dosing criteria or dosages. Therefore, statins are not yet available for general use. It is expected that statin prescriptions will gradually increase in the future.
- On page 5, line 184, the authors reported: ”The median age at the time of post-operative cardiac catherization was 59months”. In table1, this was the median age at the time of CABG, whereas the interval between CABG and angiographic follow-up was 108 months Probably something is wrong. Please check the data in the text and table or clarify.
Response: Thank you for your suggestion. The insertion point has been changed to "3.4 Latest CAG findings".
The following sentence has been added (Lines 242-243):
“The median age at the latest postoperative cardiac catheterization was 59 months since CABG.”
- The abstract might be shortened.
Response: Thank you for your guidance. The word limit is 200 words, and it is quite difficult to shorten.